# Induction of DR5-Dependent Apoptosis by PGA_2_ through ATF4-CHOP Pathway

**DOI:** 10.3390/molecules27123804

**Published:** 2022-06-13

**Authors:** Kyeong-Min Park, Ji-Young Park, Jaehyuk Pyo, Sun-Young Lee, Ho-Shik Kim

**Affiliations:** 1Department of Biomedicine and Health Sciences, College of Medicine, The Catholic University of Korea, Seoul 06591, Korea; tmxkquf1010@naver.com (K.-M.P.); jpweb@catholic.ac.kr (J.-Y.P.); biopyo37@naver.com (J.P.); 2Department of Biochemistry, College of Medicine, The Catholic University of Korea, Seoul 06591, Korea; 3Cancer Evolution Research Center, College of Medicine, The Catholic University of Korea, Seoul 06591, Korea

**Keywords:** prostaglandin A_2_, apoptosis, DR5, p53, ATF4, CHOP

## Abstract

Prostaglandin (PG) A_2_, a cyclopentenone PG, induced apoptosis in both HCT116 and HCT116 p53 −/− cells. Although PGA_2_-induced apoptosis in HCT116 cells was dependent on the p53-DR5 pathway, the mechanism underlying PGA_2_-induced apoptosis in HCT116 p53 −/− cells remains unknown. In this study, we observed that PGA_2_ caused an increase of mRNA expression of DR5 and protein expression even in HCT116 p53 −/− cells, accompanied by caspase-dependent apoptosis. Knockdown of DR5 expression by RNA interference inhibited PGA_2_-induced apoptosis in HCT116 p53 −/− cells. Parallel to the induction of apoptosis, PGA_2_ treatment upregulated expression of genes upstream of DR5 such as ATF4 and CHOP. Knockdown of CHOP prevented DR5-dependent cell death as well as the expression of DR5 protein. Furthermore, knockdown of ATF4 by RNA interference decreased both mRNA and protein levels of CHOP and DR5, thereby suppressing PGA_2_-induced cell death. Consistently, the DR5 promoter activity increased by PGA_2_ was not stimulated when the CHOP binding site in the DR5 promoter was mutated. These results collectively suggest that PGA_2_ may induce DR5-dependent apoptosis via the ATF4-CHOP pathway in HCT116 p53 null cells.

## 1. Introduction

Prostaglandin A_2_ (PGA_2_), one of cyclopentenone PGs (cycPGs), has been reported to induce apoptosis via multiple pathways depending on cell types [1]. Although it has been reported that PGA_2_-induced apoptosis is dependent on *de novo* protein synthesis and is associated with the expression of *BAX*, *SOX4*, and *c-Myc* in cervical, breast and liver cancer cell lines [2,3,4], the molecular mechanisms in detail involved in PGA_2_-induced apoptosis have not yet been completely resolved.

DR5 is a member of the tumor necrosis factor receptor (TNFR) superfamily that possesses a cytoplasmic death domain (DD) [5]. Binding of tumor necrosis factor (TNF)-related apoptosis-inducing ligand (TRAIL) to death receptors, DR4 and DR5, promotes recruitment of Fas-associated death domain (FADD) via their cytoplasmic DDs and caspase-8 to form the death-inducing signaling complex (DISC), thereby inducing extrinsic apoptosis. Expression of *DR5* can be modulated by specific transcription factors such as p53, specificity protein 1 (Sp1), nuclear factor κ-light-chain-enhancer of activated B cells (NF-κB), and C/EBP homologous protein (CHOP), whose transcriptional activities are increased by diverse triggers such as DNA damage, TNF-α, reactive oxygen species (ROS), and endoplasmic reticulum (ER) stress [6,7,8,9,10,11]. *DR5* expression can also be augmented at the level of epigenetics. Interestingly, high expression of DR5 induced by anti-cancer therapeutics through diverse mechanisms leads to activation of extrinsic apoptosis via its oligomerization, even in the absence of TRAIL [7,12,13]. After it was known that the death receptor apoptotic pathway induced by TRAIL is highly selective for tumor cells over normal cells, DR5 has been intensively pursued as a target of anti-cancer therapy development [14,15].

In a previous report, PGA_2_ was shown to induce apoptosis dependent on DR5, whose expression was stimulated by DNA-dependent protein kinase (DNA-PK)-p53 pathway in HCT116 cells [16]. However, we also observed that PGA_2_ induced apoptosis in HCT116 p53 null cells (HCT116 p53 −/−). Although the potency of PGA_2_-induced apoptosis in HCT116 p53 −/− cells was less than that in parental HCT116 cells, it implied that PGA_2_ could also induce p53-independent apoptosis. Accordingly, this study aimed to investigate the mechanism underlying p53-independent apoptosis by PGA_2_ using HCT116 p53 −/− cells which is isogenic with HCT116 cells.

## 2. Results

### 2.1. Induction of Caspase-Dependent Apoptosis in HCT116 p53 −/− Cells by PGA_2_

To determine whether PGA_2_ can induce p53-independent apoptosis, HCT116 p53 −/− cells were treated with PGA_2_ and then subjected to annexin V (AV) and propidium iodide (PI) assay. As shown in Figure 1A, AV-positive HCT116 p53 −/− cells increased significantly after PGA_2_ treatment, accompanied by activation of caspase-8, -3, and -9, and PARP1 cleavage (Figure 1B). In addition, when HCT116 p53 −/− cells were pretreated by z-VAD-Fmk, a pan-caspase inhibitor, the increase of AV-positive cells was almost completely prevented (Figure 1C). Thus, these findings indicate that PGA_2_ induces caspase-dependent and p53-independent apoptosis in HCT116 p53 −/− cells.

### 2.2. Induction of DR5-Dependent Apoptosis in HCT116 p53 −/− Cells by PGA_2_

Since PGA_2_-induced apoptosis is usually dependent on *de novo* protein synthesis and is reported to be dependent on the expression of *DR5* gene in HCT116 cells [16], we first analyzed the expression change of *DR5*. As shown in Figure 2A, the mRNA and protein levels of *DR5* significantly all increased, implying the causative effect of DR5 in this apoptosis. Consistently with this implication, knockdown of DR5 using small interfering RNA (siRNA) suppressed PGA_2_-induced increase of AV-positive cells and the cleavage of PARP-1 as well (Figure 2B,C). These results collectively suggest that PGA_2_-induced apoptosis in HCT116 p53 −/− cells is dependent on *DR5* expression.

### 2.3. Involvement of CHOP in PGA_2_-Induced Apoptosis and Expression of DR5

It is generally known that the most important transcription factors that enhance the transcription of *DR5* are Sp1, p53, CHOP, NF-κB, and yin-yang 1 (YY1) [6]. Since the cell line used in this study is *p53* deleted, and it was reported that cycPGs accumulate in the endoplasmic reticulum (ER), thereby inducing the expression of ER stress-related genes [17,18], we expected that CHOP might be the key transcription factor to stimulate *DR5* expression in PGA_2_-treated HCT116 p53 −/− cells. And as expected, *CHOP* expression was up-regulated by PGA_2_ at the level of both mRNA and protein along with an increase of DR5 protein (Figure 3A), and knockdown of CHOP prevented the increase of *DR5* mRNA and protein expression (Figure 3B). Besides, whereas PGA_2_ stimulated the *DR5* promoter activity, *DR5* promoter with the CHOP binding site mutated was not responded to by PGA_2_, indicating that PGA_2_-induced *DR5* expression is mediated by CHOP at the level of transcription (Figure 3C). Consistently with the expression change, PGA_2_-induced apoptosis was suppressed by knockdown of CHOP as shown in AV/PI and immunoblot analyses (Figure 3D,E). Taken together, these data suggest that PGA_2_ induces expression of *CHOP*, which in turn stimulates the transcription of *DR5*, resulting in apoptosis in HCT116 p53 −/− cells.

### 2.4. The Role of ATF4 in PGA_2_-Induced Apoptosis and Activation of CHOP-DR5 Pathway

Activating transcription factor (ATF) 3 and ATF4 are stress-responsive proteins that can conduce CHOP transcription in response to various stimuli. Moreover, it was reported that ATF4-ATF3-CHOP cascade induced apoptosis in human pulmonary cancer and leukemia cells [19,20]. Thus, to identify the upstream regulators of CHOP, we analyzed the expression of ATF3 and ATF4. As shown in Figure 4A,B, PGA_2_ increased the expression of ATF3 and 4 at the levels of mRNA and protein, implying the ATF3 and 4-mediated expression of *CHOP* and *DR5*. Although PGA_2_ increased the expression of ATF3 with a parallel level to that of ATF4, only knockdown of ATF4 inhibited the induction of apoptosis (Figure 4C and Appendix A). Furthermore, ATF4 siRNA suppressed the PGA_2_-induced increase of CHOP and DR5 proteins (Figure 4D), whereas ATF3 siRNA did not affect it. Collectively, therefore, it can be concluded that PGA_2_ induces apoptosis by activating the ATF4-DR5-CHOP pathway.

## 3. Discussion

In a previous report, PGA_2_ was shown to activate DR5-dependent apoptosis in HCT116 cells containing wild-type p53 [16]. In HCT116 cells, PGA_2_ induced phosphorylation of both histone H2AX, a hallmark of DNA double-strand breaks, and DNA-PK, which in turn phosphorylated p53 with an elevation of its transcriptional activity, resulting in upregulation of *DR5* expression and finally induction of apoptosis. So, apoptosis activated by PGA_2_ in HCT116 cells seemed to be dependent on the p53-induced expression of *DR5*. However, it turned out that PGA_2_ could trigger DR5-dependent apoptosis even in the absence of p53 in isogenic HCT116 cells with p53 genetically deleted (HCT116 p53 −/− cells) in this study. In HCT116 p53 −/− cells treated with PGA_2_, histone H2AX was also phosphorylated, indicating that PGA_2_ damaged DNA as in parental HCT116 cells (Appendix A). However, NU7441 and KU5593, chemical inhibitors of DNA-PK and ATM activities, respectively, did not prevent either PGA_2_-induced expression of *DR5* (Appendix A). Collectively, these data suggest that PGA_2_-induced DNA damage leading to *DR5* expression should be via the activity of p53 but not DNA-PK *per se* in HCT116 cells, and thus, in the absence of p53 like HCT116 p53 −/− cells, other pathways might be activated to increase *DR5* expression. Interestingly, in annexin V assays, since PGA_2_ also increased PI staining albeit subtlety, PGA_2_ might potentially induce necrosis in HCT116 p53 −/− cells, implying the activation of various cell death pathways by PGA_2_ in the absence of p53.

PGA_2_, a cycPG, is formed by non-enzymatic dehydration of PGE_2_. While PGE_2_ exerts its function by binding to the PGE_2_ receptors which are G-protein coupled receptors in the plasma membrane, PGA_2_, like other cycPGs, enters cells and accumulates in subcellular organelles such as nuclei and mitochondria, where it interacts with diverse molecules such as DNA and cellular proteins including transcription factors [21]. For example, in the cytosol, PGA_2_ binds transcriptional and signaling regulators such as KEAP1 and IKK-β subunit of IKK [22,23]. KEAP1 binding of PGA_2_ induces Nrf2 activation, and IKK-β subunit binding prevents NF-κB activation and hence, repression of LPS-induced inflammatory response. As mentioned above, accumulated PGA_2_ in nuclei might damage DNA, which leads to DR5-dependent apoptosis. PGA_2_ also directly interacts with mitochondria, which causes mitochondrial outer membrane permeabilization through which cytochrome c releases into cytosol resulting in the initiation of the intrinsic apoptosis [24]. Interestingly, it was reported that cycPGs, including PGA_2_, localize the endoplasmic reticulum (ER), and PGJ_2_, another cycPG, could stimulate transcription of BiP, an ER-stress response gene, through an unfolded protein response element [17,18]. Like PGJ_2_, PGA_2_ enhanced the expression of BiP at the levels of mRNA and protein (Appendix A). Notably, PGA_2_ also increased the expression of ER-stress response genes such as *ATF3*, *ATF4*, and *CHOP*, implying that the ER-stress response might be involved in PGA_2_-induced *DR5* expression in HCT116 p53 −/− cells. Moreover, as expected, phosphorylation of eIF2α and PERK was increased, accompanied by the expression of *ATF3*, *ATF4*, and *CHOP* (Appendix A). So, it can be speculated that PGA_2_ may induce the ER-stress response leading to activation of ATF4–CHOP–DR5 pathway, although how PGA_2_ induces the ER stress response was not investigated in this study.

To expand the mechanism of this study to colon cancer cells, we also analyzed the apoptosis induction activity of PGA_2_ in SW620 cells of which p53 is mutated. However, as shown in Appendix A, PGA_2_ did not induce apoptosis up to 48 h in SW620 cells. So, it can be summarized that PGA_2_ induces fulminant apoptosis in HCT116 p53 WT cells, slight apoptosis in HCT116 p53 −/− cells, and no apoptosis in p53-mutated SW620 cells. Although we did not analyze many colon cancer cell lines, these findings suggest that p53 plays a critical role in apoptosis induction by PGA_2_, but PGA_2_ induction of apoptosis in colon cancer cells is not determined only by p53 genetic status. Moreover, depending on the cellular context, PGA_2_ may have the potential to induce apoptosis even in p53-mutated colon cancer cells, as shown in this study that PGA_2_ induces apoptosis via ER-stress–ATF4–CHOP–DR5 pathway. It can also be expected that PGA_2_ may induce apoptosis through various pathways in a p53-independent manner.

TRAIL is a potent inducer of apoptosis selectively in cancer cells. TRAIL induces apoptosis via DR4 and DR5 [15,25]. However, some cancers develop TRAIL resistance by downregulating the expression of death receptors or upregulating decoy receptors which hinders the binding of TRAIL to death receptors [26]. Thus, molecules that increase the expression of death receptors may have the potential to overcome TRAIL resistance [27,28]. As shown in Appendix A, PGA_2_ pretreatment significantly potentiated TRAIL-induced apoptosis in HCT116 p53 −/− cells as well as HCT116 cells. Although the effect of PGA_2_ in TRAIL-resistant cells was not examined yet, this finding may shed a sound effect of PGA_2_ on the recovery of TRAIL resistance.

It is firmly established that tumor suppressor *p53* is the most important gene in anti-carcinogenesis and induction of apoptosis in cancer cells [29]. In fact, most anti-cancer therapeutics exert their effects via the activity of p53. But, unfortunately, almost half of all cancers contain a mutation of p53, which could contribute to the development of resistance to anti-cancer therapeutics [30]. Therefore, the development of an anti-cancer agent that induces both p53-dependent and -independent apoptosis in cancer cells may be of high value in cancer patient treatment. Therefore, expression of *DR5* induced by PGA_2_ via p53 and CHOP in cancer cells containing wild-type p53 and mutant p53, respectively, may have the potential to enhance the anti-cancer therapeutic effect of TRAIL.

## 4. Materials and Methods

### 4.1. Cell Culture

Human colon cancer HCT116 p53 −/− cells were maintained in RPMI 1640 supplemented with 10% fetal bovine serum (Hyclone, Logan, UT, USA), 100 units/mL penicillin (Hyclone), and glutamate (Invitrogen, Carlsbad, CA, USA). The cells were incubated in a humidified atmosphere of 5% CO_2_ at 37 °C.

### 4.2. Chemicals

PGA_2_ was obtained from Enzo Life Sciences, Inc. (Farmingdale, NY, USA). Z-VAD-Fmk was purchased from Tocris Bioscience (Bristol, UK). All reagents used in this study were of molecular biology or cell culture tested grade.

### 4.3. Transfection with Small Interfering RNA (siRNA)

According to the manufacturer’s instruction, siRNA transfection was performed with Lipofectamine RNAiMAX (Invitrogen). Briefly, the mixture containing siRNA, Opti-MEM (Life Technologies, Carlsbad, CA, USA), and Lipofectamine RNAiMAX was incubated for 15 min at room temperature (RT), then dropped to the cells and incubated for 24 h. Subsequently, the cells were treated with PGA_2_ for further analysis. DR5 siRNA and ATF4 siRNA were purchased from Bioneer (Daejeon, Korea). CHOP siRNA and ATF3 siRNA were obtained from Invitrogen and Santa Cruz Biotechnology (Santa Cruz, CA, USA), respectively.

### 4.4. Apoptosis Assay

The phosphatidylserine on the extracellular side of cell membranes and propidium iodide to detect necrotic nuclei using an Annexin V-FITC/PI cell apoptosis detection kit (BD Pharmingen, San Diego, CA, USA) according to the manufacturer’s instruction. Then, the fluorescence of 10,000 stained cells was measured on FACS Canto II (BD Biosciences, San Jose, CA, USA) and analyzed using BD FACS Diva program.

### 4.5. Immunoblot Analysis and Antibodies

For immunoblot analysis, proteins were extracted from PGA_2_-treated cells. Cells were lysed in RIPA buffer (Cell signaling Technology, Boston, MA, USA) containing cocktails of protease inhibitors (Roche, Basel, Switzerland) and phosphatase inhibitors (FIVE photon Biochemicals, San Diego, CA, USA). All antibodies were commercially available as follows. Rabbit anti-cleaved PARP1 (c-PARP1), anti-ATF3, anti-ATF4, anti-DR5 (death receptor 5), anti-caspase-3, anti-cleaved caspase-3, anti-caspase-9, and mouse anti-CHOP were purchased from Cell Signaling Technology (Boston, MA, USA). Chicken anti-GAPDH antibodies were obtained from Merck Millipore Korea (Seoul, Korea), respectively. Peroxidase-conjugated antibodies (HRP-conjugated anti-rabbit or -mouse IgG) were from Sigma-Aldrich Inc., and KPL (Gaithersburg, MD, USA, HRP-conjugated anti-chicken IgG).

### 4.6. Quantitative Real-Time PCR (qPCR)

Total RNAs were extracted with RNAiso plus reagent (Takara Korea Biomedical Inc., Seoul, Korea). First-strand cDNA was synthesized from total RNA using PrimeScript^TM^ RT reagent Kit (Takara Korea Biomedical Inc.). First-strand cDNA was then amplified by specific primers against various genes using SYBR FAST qPCR Kit (Nanohelix, Daejeon, Korea) on ABI 7300 Real-Time PCR System (Applied Biosystems, Carlsbad, CA, USA). GAPDH mRNA level normalized each mRNA level, and relative changes among samples were calculated using the ΔΔCt method [31].

### 4.7. Luciferase Reporter Gene Assay

The activity of the human pGL3-DR5(-605)-Luc promoter and pGL3-DR5 (mt. CHOP)-Luc was measured by dual luciferase-reporter gene assay. pGL3-DR5(-605)-Luc promoter contains the CHOP binding site in DR5 promoter sequence, and DR5 (mt. CHOP) promoter-luciferase plasmid contains a mutated CHOP binding site. All promoter-luciferase plasmids were generously donated by Dr. Taeg Kyu Kwon (Keimyung University, Daegu, Korea). To measure the activity of DR5 promoter, cells were transfected with 0.2 μg of pGL3-DR5(-605)-Luc or pGL3-DR5(mt. CHOP)-Luc, together with 0.02 μg of Renilla luciferase using Xtreme Gene 9 (Roche Diagnostics, Basel, Switzerland) for 24 h, and then treated with PGA_2_. At the indicated times post-treatment, Firefly and Renilla luciferase activities were determined using Dual-Luciferase kit (Promega, Fitchburg, WI, USA), and data were expressed as relative luciferase activity (RLA) of three independent experiments performed in triplicate. Renilla luciferase activity was used to normalize transfection efficiency, and normalized firefly luciferase activity of non-treated cells was set to RLA 1 fold.

### 4.8. Statistical Analysis

All data in this study are expressed as the means ± standard error of the mean (SEM) obtained from three independent experiments performed in triplicate. Statistical analysis was performed using Student’s two-tailed *t*-test for paired data. *p* values of the results were indicated in each figure.

## Figures and Tables

**Figure 1 molecules-27-03804-f001:**
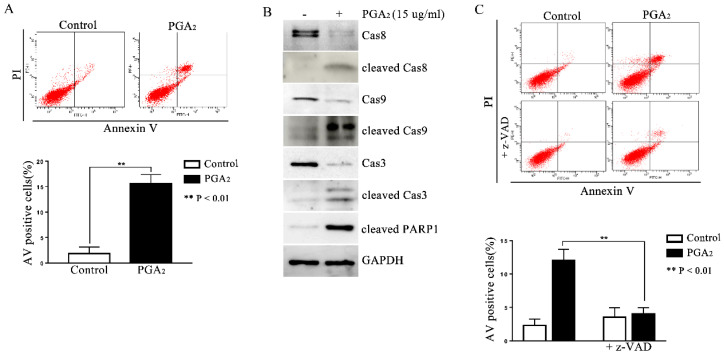
Induction of caspase-dependent apoptosis in HCT116 p53 −/− cells by PGA_2_. (**A**) HCT116 p53 −/− cells were treated by PGA_2_ (15 μg/mL) for 30 h. Cells were then stained with annexin V and propidium iodide, which were subjected to flow cytometric analysis. A representative image of three independent annexin V assays was shown (upper panel), and the quantitative result was presented as mean ± SEM (lower panel). (**B**) Whole cell lysates (WCL) of HCT116 p53 −/− cells treated the same as described in (**A**) were subjected to immunoblot analysis against procaspase-8 (Cas8), cleaved caspase-8 (cleaved Cas8), procaspase-9 (Cas9), cleaved caspase-9 (cleaved Cas9), procaspase-3 (Cas3), cleaved caspase-3 (cleaved Cas3), and cleaved PARP1 (cleaved PARP1). GADPH was used as an internal reference protein for normalization. Densitometric measurement of three independent immunoblot analyses was represented as mean ± SEM in Appendix A. (**C**) HCT116 p53 −/− cells incubated in the presence of vehicle or z-VAD-Fmk for 1 h were treated with vehicle or 15 μg/mL of PGA_2_ for another 30 h. Cells were then analyzed by annexin V assay. The picture is representative of three independent experiments (upper panel), and their quantitative results were presented as mean ± SEM (lower panel).

**Figure 2 molecules-27-03804-f002:**
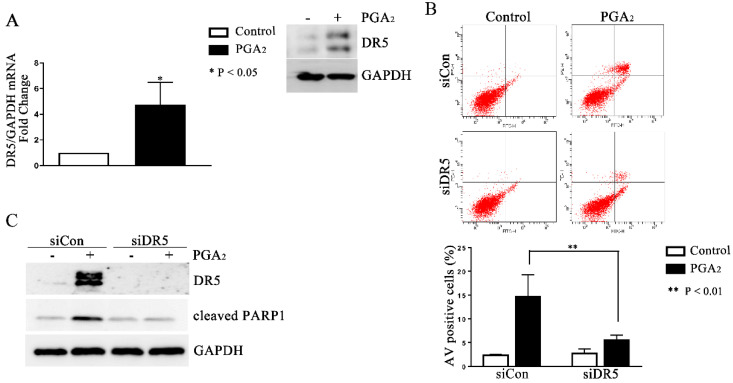
Induction of DR5-dependent apoptosis in HCT116 p53 −/− cells by PGA_2_. (**A**) Total cellular RNA extracted from HCT116 p53 −/− cells treated with PGA_2_ (15 μg/mL) for 30 h were subjected to qPCR against indicated genes using GADPH as an internal reference gene for normalization. (**B**) HCT116 p53 −/− cells were transfected with scrambled RNA (siCon) or siRNA targeting *DR5* (siDR5) for 24 h and incubated in the presence of vehicle or PGA_2_ (15 μg/mL) for an additional 30 h. Cells were then subjected to annexin V assay. The result is representative of three independent experiments (upper panel), and their quantitative result was presented as mean ± SEM (lower panel). (**C**) Cells treated the same as described in (**B**) were subjected to immunoblot analysis against DR5 and cleaved PARP-1 using GAPDH as an internal reference protein. Densitometric measurement of three independent immunoblot analyses was represented as mean ± SEM in Appendix A.

**Figure 3 molecules-27-03804-f003:**
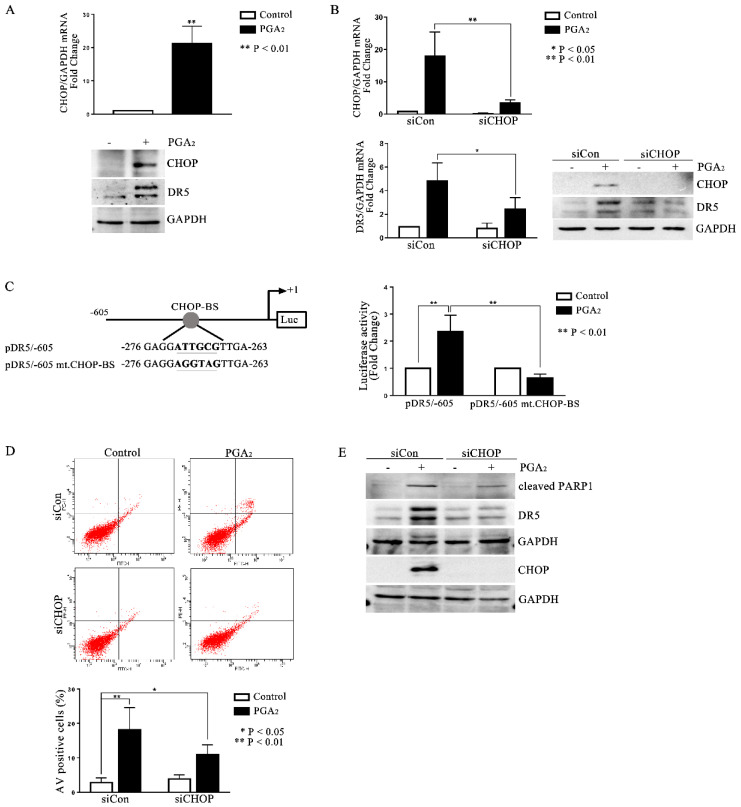
Involvement of CHOP in PGA_2_-induced apoptosis and expression of *DR5*. (**A**) HCT116 p53 −/− cells were treated with vehicle or PGA_2_ (15 μg/mL) for 30 h. Cells were then subjected to qPCR against *CHOP* gene using *GADPH* as an internal reference gene for normalization (upper) or immunoblot analysis against indicated proteins with GAPDH as the normalizer (lower). (**B**) HCT116 p53 −/− cells were transfected with scrambled RNA or siRNA targeting *CHOP* for 24 h and incubated in the presence of vehicle or PGA_2_ (15 μg/mL) for an additional 30 h. Cells were then subjected to qPCR or immunoblot analysis against CHOP and DR5 using GAPDH as an internal reference protein or gene. (**C**) *DR5* promoter or mutant *DR5* promoter of which CHOP-binding site was changed, was transfected into HCT116 p53 −/− cells along with Renilla luciferase for 24 h, and HCT116 p53 −/− cells were then treated with vehicle or PGA_2_. At 30 h post-treatment, firefly luciferase activity of *DR5* promoter was measured by the Dual-luciferase assay method using Renilla luciferase activity as the normalizer. (**D**) HCT116 p53 −/− cells were transfected with scrambled RNA (siCon) or siRNA targeting *CHOP* (siCHOP) for 24 h and incubated in the presence of vehicle or PGA_2_ (15 μg/mL) for an additional 30 h. Cells were then subjected to annexin V assay. The result is representative of three independent experiments (upper), and their quantitative result was presented as mean ± SEM (lower). (**E**) WCL of HCT116 p53 −/− cells treated the same as described in (**D**) were subjected to immunoblot analysis against indicated proteins using GAPDH as an internal reference protein. Densitometric measurement of three independent immunoblot analyses was represented as mean ± SEM in Appendix A.

**Figure 4 molecules-27-03804-f004:**
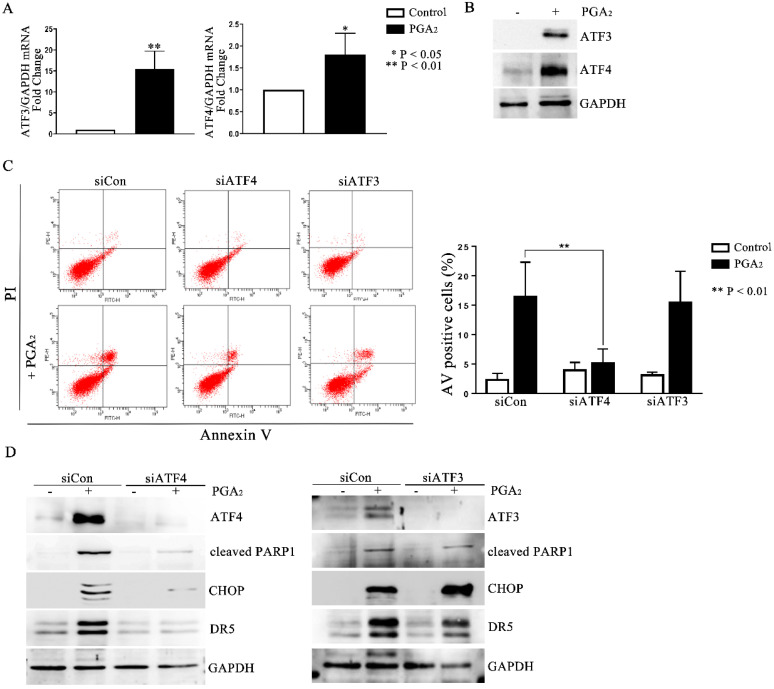
The role of ATF4 in the PGA_2_-induced apoptosis and activation of the CHOP-DR5 pathway. (**A**) Total cellular RNA extracted from HCT116 p53 −/− cells treated with PGA_2_ (15 μg/mL) for 30 h were subjected to qPCR against *ATF3* or *ATF4* using *GADPH* as an internal reference gene for normalization. (**B**) WCL of cells treated the same as described in (**A**) were subjected to immunoblot analysis against ATF3 and ATF4 using GAPDH as the normalizer. (**C**) HCT116 p53 −/− cells were transfected with scrambled RNA (siCon) or siRNA targeting ATF3 (siATF3) or ATF4 (siATF4) for 24 h and incubated in the presence of vehicle or PGA_2_ (15 μg/mL) for an additional 30 h. Cells were then subjected to annexin V assay. The result is representative of three independent experiments (left panel), and their quantitative result was presented as mean ± SEM (right panel). (**D**) Cells treated the same as described in (**C**) were subjected to immunoblot analysis against indicated proteins using GAPDH as an internal reference protein. Densitometric measurement of three independent immunoblot analyses was represented as mean ± SEM in Appendix A.

## Data Availability

Not applicable.

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
