# Peer review of "Induction of DR5-Dependent Apoptosis by PGA2 through ATF4-CHOP Pathway"

_molecules, 2022, doi:10.3390/molecules27123804_

Round 1
Reviewer 1 Report
The manuscript is well written. It is clear and well described and focused.
I think that the only limit of this study is that the authors use just one cell line. Could the authors implement their data by using other colon cell lines? Why the authors choose the HCT116 cell system? They tested also other tumor cell lines? It should be interesting also to know what happens in other tumors, considering the p53 status as well. I think that the fact that the authors focus all their manuscript on just one cell system is too less informative but I think that if they consider the opportunity to expand their results as I suggested previously, I’m sure that the manuscript could have a major impact.
Anyway, I appreciate a lot the fluidity of the text, the figure are good and above all the results and discussion are very clear and detailed.
Just a little comment on the Figure 4C, in which there are any indication on y axis (is referred for?).
Author Response
The manuscript is well written. It is clear and well described and focused.
I sincerely appreciate your favorable assessment of our manuscript.
I think that the only limit of this study is that the authors use just one cell line. Could the authors implement their data by using other colon cell lines? Why the authors choose the HCT116 cell system? They tested also other tumor cell lines? It should be interesting also to know what happens in other tumors, considering the p53 status as well. I think that the fact that the authors focus all their manuscript on just one cell system is too less informative but I think that if they consider the opportunity to expand their results as I suggested previously, I’m sure that the manuscript could have a major impact.
I appreciate your impressive questions. Your comments are the most critical issues of this manuscript and are actually what I want to know. We also tested the apoptosis-inducing effect of PGA2 in SW620 cells of which p53 is mutated. As shown in the supplementary Figure S6, PGA2 did not induce apoptosis up to 48 hours in SW620 cells. So, it can be summarized that PGA2 induces fulminant apoptosis in HCT116 p53 WT cells, slight apoptosis in HCT116 p53 null cells, and no apoptosis in p53-mutated SW620 cells. Moreover, I thought that the most important message is that PGA2 induction of apoptosis in colon cancer cells is not determined only by p53 genetic status. However, since we still wanted to know the precise role of p53 in the PGA2-induced apoptosis of colon cancer cells, we used isogenic HCT116 and HCT116 p53 null cells.
I absolutely agree with you that this study's limitation is that only one cell line was used. So, I described the limitation of this study as follows in the Discussion section.
To expand the mechanism of this study to colon cancer cells, we also analyzed the apoptosis induction activity of PGA2 in SW620 cells of which p53 is mutated. However, as shown in Figure S6, PGA2 did not induce apoptosis up to 48 hours in SW620 cells. So, it can be summarized that PGA2 induces fulminant apoptosis in HCT116 p53 WT cells, slight apoptosis in HCT116 p53 -/- cells, and no apoptosis in p53-mutated SW620 cells. Although we did not analyze many colon cancer cell lines, these findings suggest that p53 plays a critical role in apoptosis induction by PGA2, but PGA2 induction of apoptosis in colon cancer cells is not determined only by p53 genetic status. Moreover, depending on the cellular context, PGA2 may have the potential to induce apoptosis even in p53-mutated colon cancer cells, as shown in this study that PGA2 induces apoptosis via ER-stress – ATF4 – CHOP – DR5 pathway. It can also be expected that PGA2 may induce apoptosis through various pathways in a p53-independent manner.
Anyway, I appreciate a lot the fluidity of the text, the figures are good and above all the results and discussion are very clear and detailed.
Thank you so much again for your beneficial evaluation of our manuscript.
Just a little comment on the Figure 4C, in which there are any indication on y axis (is referred for?).
Thank you so much for pointing it out The Y-axis of Figure 4C is % of AV-positive cells. I added an indication (AV positive cells (%)) on the y-axis in the revised version.
Reviewer 2 Report
In this work, Park et al evaluate the mechanistic insights regulating PGA2-induced apoptosis in HCT166 cells harboring p53 mutation. They find that PGA2 may induce DR5-dependent apoptosis via the ATF4-CHOP pathway 24 in HCT116 p53 null cells. The article is interesting and well-written. Nevertheless, some major and minor issues should be conducted previous to final acceptance of the article in Molecules.
Major comments
1) In page 2, line 65, the authors claim than caspases 8 and 9 are activated, however, in Figure 1B Western blots for cleaved caspases 8 and 9 are not showed.
2) It would be useful also to show a graph depicting the growth curve of cells upon PGA2 administration and the subsequent “rescue experiments”
3) In general, all western-blot assays should be accompanied by graphs depicting protein levels quantitation (in both main and supplementary figures).
4) Have the authors any information about PGA2-mediated apoptosis through ATF4-DR5-150 CHOP pathway is exclusive of p53-/- HCT116 cells? Could also participate in PGA2-induced apoptosis in HCT116 p53 WT cells?
Minor comments
1) Please indicate how many cells are counted in Flow Cytometry studies.
2) In Figure S2, why the authors evaluate PARP expression rather than cleaved-PARP.
3) The authors should improve the quality of pPERK WB in Figure S4. In addition, expression levels of total PERK and total eIF2a should be measured and added to this figure.
Author Response
In this work, Park et al evaluate the mechanistic insights regulating PGA2-induced apoptosis in HCT166 cells harboring p53 mutation. They find that PGA2 may induce DR5-dependent apoptosis via the ATF4-CHOP pathway 24 in HCT116 p53 null cells. The article is interesting and well-written. Nevertheless, some major and minor issues should be conducted previous to final acceptance of the article in Molecules.
I sincerely appreciate your valuable assessment of our manuscript and your critical pointing. Following your comments, we tried out best to revise our manuscript.
Major comments
1) In page 2, line 65, the authors claim than caspases 8 and 9 are activated, however, in Figure 1B Western blots for cleaved caspases 8 and 9 are not showed.
Thank you so much for your pointing it out. We performed western blot analysis with anti-cleaved caspase-9 and anti-cleaved caspase-8 and inserted its results in Figure 1B.
2) It would be useful also to show a graph depicting the growth curve of cells upon PGA2 administration and the subsequent “rescue experiments”
Thank you so much for your very helpful suggestion. We have contained the growth curve of HCT116 p53 null cell lines treated by PGA2 in a previous report (supplementary Figure S14, Biomolecules 2020,10,492). The growth was inhibited in HCT116 p53 null cells by PGA2, the potency of which was less than that in HCT116 p53 wild-type cells. The growth inhibition by PGA2 appears to be consistent with the apoptosis induction activity of PGA2. Therefore, the growth inhibitory effect of PGA2 in HCT116 cells may be via induction of apoptosis.
3) In general, all western-blot assays should be accompanied by graphs depicting protein levels quantitation (in both main and supplementary figures).
I deeply appreciate your pointing out. We performed densitometric analysis and contained the densitometric results as a supplementary file (densitometry.pdf) according to your suggestion.
4) Have the authors any information about PGA2-mediated apoptosis through ATF4-DR5-150 CHOP pathway is exclusive of p53-/- HCT116 cells? Could also participate in PGA2-induced apoptosis in HCT116 p53 WT cells?
I sincerely appreciate your critical questions. Your question is impressive and seems to be very challenging. Since phosphorylation of eIF2a and expression of BiP was not induced in HCT116 p53 WT cells, I do not expect the possible involvement of ER stress-ATF4-CHOP-DR5 pathway in PGA2-induced apoptosis of HCT116 p53 WT cells and thus did not analyze the effect of ATF4 in apoptosis of HCT116 p53 WT cells. However, since the transcription of ATF4 could be modulated by multiple signaling pathways, including mTORC1 and deficiency of base excision repair [1,2], the role of ATF4 in HCT116 p53 WT cells is worth to be investigated in the future. Therefore, I would like to express my gratitude for your insightful comments.
[1] Torrence ME, MacArthur MR, Hosios AM, Valvezan AJ, Asara JM, Mitchell JR, Manning BD. The mTORC1-mediated activation of ATF4 promotes protein and glutathione synthesis downstream of growth signals. Elife. 2021;10:e63326. doi: 10.7554/eLife.63326.
[2] Markkanen E, Fischer R, Ledentcova M, Kessler BM, Dianov GL. Cells deficient in base-excision repair reveal cancer hallmarks originating from adjustments to genetic instability. Nucleic Acids Res. 2015;43:3667–79.
Minor comments
1) Please indicate how many cells are counted in Flow Cytometry studies.
Thank you so much for your pointing out. Ten thousand cells were counted in every Flow Cytometric Analysis. I indicated cell numbers in the Materials and Methods section as follows.
Then, the fluorescence of 10,000 stained cells was measured on FACS Canto II (BD Biosciences, San Jose, CA, USA) and analyzed using BD FACS Diva program.
2) In Figure S2, why the authors evaluate PARP expression rather than cleaved-PARP.
We ran out of anti-cleaved-PARP1, and unfortunately, my research budget was almost consumed. Moreover, the foreign exchange rate went up high, so I could not afford to buy a new anti-cleaved-PARP1. Instead, my student should do western blot analyses with anti-PARP1 that we had bought before. Anti-PARP1 detects both intact PARP1 and a high molecular weight form of cleaved PARP1, which is also suitable for assessing apoptosis induction. So, the conclusion of this study is still consistent even when using anti-PARP1. Nonetheless, I feel sorry that we used anti-PARP1 and made you confused.
3) The authors should improve the quality of pPERK WB in Figure S4. In addition, expression levels of total PERK and total eIF2a should be measured and added to this figure.
Following your suggestion, we performed western blot analysis against phospho-PERK, total PERK, and total eIF2a. Fortunately, we obtained more explicit images than those in the original version. Accordingly, we replaced and inserted new images in Figure S4 and performed densitometric analysis which shows a dose-dependent increase in phosphorylation of both PERK and eIF2a. The improvement of the quality of these data is owing to your suggestion. So, we would like to express our sincere gratitude to you.
Reviewer 3 Report
The manuscript is an off shoot from a work previously done by the authors. They explain how a small number of cells may be killed by PGA2 in p53 independent manner. The flow of logic is sound. However, I do have concerns on some of the data that is presented.
- Figure 1 A and all FACS figures : The increase in fluorescence is primarily in the PE channel (PI). Their previous paper on this topic shows a different sort of shift upon apoptosis (more of An-V and similar PI). How do the authors justify the difference between the two manuscripts? (Please also label the Axis for AV and PI properly).
- Figure 1C: Please recalculate if there is indeed a >10% AV positivie cells in PAG2 treated condition. The image does not show the same.
- Figure 4C: The representative cytometry data doesnt seem to represent the bar graphs. It is hard to visually see the difference between siATF4+PGA2 and siATF3+pGA2.
Overall a very concisely presented manuscript with important implications.
Author Response
The manuscript is an off shoot from a work previously done by the authors. They explain how a small number of cells may be killed by PGA2 in p53 independent manner. The flow of logic is sound. However, I do have concerns on some of the data that is presented.
I sincerely appreciate your generous assessment of our manuscript. According to your suggestions which I believe, contribute to the completeness of our manuscript, we revised our manuscript.
1. Figure 1 A and all FACS figures : The increase in fluorescence is primarily in the PE channel (PI). Their previous paper on this topic shows a different sort of shift upon apoptosis (more of An-V and similar PI). How do the authors justify the difference between the two manuscripts? (Please also label the Axis for AV and PI properly).
Thank you so much for your pointing out. You pointed out a very critical point. Since AV and PI are the two essential parameters that determine the types of cell death, that is, apoptosis and necrosis, we tried our best to get accurate images of HCT116 and HCT116 p53 null cells. However, as you pointed out, we found that PI staining intensity is more substantial in HCT116 p53 null cells than that in HCT116 cells. We tried all the ways we could do, including compensation between FITC and PE and adjustment of PI concentration, but in vain.
Nonetheless, the increase of annexin V-positive cells was consistent since AV positive % in dual measurement of FITC and PE was the same as in the histogram of AV. Therefore, the conclusion of the annexin v assay seems to be accurate. Although I don’t know the reason why HCT116 p53 null cells show the higher intensity of PI, I just assume that deletion of p53 might affect the integrity of cytoplasmic and nuclear membranes, which leads to higher PI staining of DNA at the basal state. In addition, PGA2 might induce subtle necrosis in HCT116 p53 null cells. The possible necrosis induction by PGA2 should be investigated soon following this manuscript's completion.
To justify the difference, I inserted the description as follows in the Results section.
Interestingly, in annexin V assays, since PGA2 also increased PI staining albeit subtlely, PGA2 might potentially induce necrosis in HCT116 p53 -/- cells, implying the activation of various cell death pathways by PGA2 in the absence of p53.
2. Figure 1C: Please recalculate if there is indeed a >10% AV positivie cells in PAG2 treated condition. The image does not show the same.
Thank you so much for the pointing out. Following your suggestion, I replaced the image with another one in Figure 1C which shows apoptosis more explicit than the previous one.
3. Figure 4C: The representative cytometry data doesnt seem to represent the bar graphs. It is hard to visually see the difference between siATF4+PGA2 and siATF3+pGA2.
Thank you very much for your careful inspection. I agree with you that this image doesn’t seem to be the representative one. So we analyzed again. As shown in Figure 4C, we think that the new images show the difference between siATF4 + PGA2 and siATF3 + PGA2 more clearly than the original one. Moreover, the histogram of FITC-Annexin V shows the difference once again. So, we contained the histogram in supplementary Figure S7.
Overall a very concisely presented manuscript with important implications.
I sincerely appreciate your favorable evaluation of our manuscript.
Round 2
Reviewer 1 Report
The authors answered the questions I did well and I appreciate that they included another cell line.
Author Response
The authors answered the questions I did well and I appreciate that they included another cell line.
I sincerely appreciate your beneficial evaluation of our revised manuscript. Your comments and questions were very insightful. Thus, I believe that our manuscript improved by revisions following your advice.
Once again, I appreciate your thoughtful review of our manuscript.
Reviewer 2 Report
The authors have improved the quality of the manuscript in the revised version of the manuscript. However, there is one question that should be clarified and corrected.
The densitometry graphs lack the error bars. Is this a mistake, or by contrast, this information indicates that only one sample has been measured? If that the case, the authors must increase the number of samples to be analyzed (at least 3), as they do in the qPCR studies.
Could the authors improve the quality of cleaved-caspase 8 in Figure 1B? Honestly, it is difficult to concede that signal as specific? Could the authors upload the original blot of this western-blot.
Author Response
Attached please find a pdf file (Response to Reviewer 2)
The authors have improved the quality of the manuscript in the revised version of the manuscript. However, there is one question that should be clarified and corrected.
I sincerely appreciate your valuable evaluation of our revised manuscript. Following your questions, we tried our best to revise the manuscript again.
The densitometry graphs lack the error bars. Is this a mistake, or by contrast, this information indicates that only one sample has been measured? If that the case, the authors must increase the number of samples to be analyzed (at least 3), as they do in the qPCR studies.
Thank you very much for your pointing it out. Since the densitomery data in the previous version is the measurement of the representative immunoblot analysis image shown in the Figures, they lack the error bars. According to your suggestion, we analyzed two more images of each figure and thus, we could add the error bars. The densitometric results of three immunoblot images were presented as mean ± SEM. I replaced previous densitometric graphs with new graphs containing the error bars in the revised version and revised the supplementary figure legends as well.
Could the authors improve the quality of cleaved-caspase 8 in Figure 1B? Honestly, it is difficult to concede that signal as specific? Could the authors upload the original blot of this western-blot.
Thank you so much for raising critical issues. I also agree with you that the signal of cleaved caspase-8 is not clear. Nonetheless, I could not help but contain the image since I don’t have enough time to perform the immunoblot analysis again. Even after resubmitting the revised version, I have struggled to get more specific image. We used anti-cleaved caspase-8 from cell signaling technology (CST, Cat NO. #CST-9748). Because this antibody catches p10 cleaved form of caspase-8, which is a very small molecular weight, we used a gradient polyacrylamide gel, 4 – 20% and 5% bovine serum albumin (BSA) solution for blocking a membrane. To dilute the antibody, we used 5% BSA in TBS-T and incubated the antibody overnight at 4℃ to increase the specific hybridization and the protein band intensity. And finally, we get the image as follows, which seems more specific than before (please see the pdf file attached).
We also performed an immunoblot analysis against BID, a substrate of active caspase-8, to verify the activation of caspase-8. As shown below, the intensity of BID was decreased by PGA2 treatment. Moreover, the decrease of BID was prevented by z-IETD-Fmk pretreatment, an inhibitor of active caspase-8 (please see the pdf file attached).
Therefore, these results confirm that PGA2 activates caspase-8 in HCT116 p53 -/- cells, and the indicated protein band is the cleaved caspase-8. I replaced the previous cleaved caspase-8 image in Figure 1B with this new one.
Once again, I sincerely appreciate your critical questions.
